# ART-FREE GENERATIVE MODELS: EXPLORING ART CREATION WITHOUT PRIOR ARTISTIC KNOWLEDGE

## ABSTRACT

In this work, we explore the question: "How much prior art knowledge is needed to create art?" To investigate this, we propose a text-to-image generation model trained without access to art-related content. We then introduce a simple yet effective method to learn an art adaptor using only a few examples of selected artistic styles. Our experiments show that art generated using our method is perceived by users as comparable to art produced by models trained on large, art-rich datasets. Finally, through data attribution techniques, we illustrate how examples from both artistic and non-artistic datasets contributed to the creation of new artistic styles.

## 1 INTRODUCTION

Is exposure to art truly necessary for creating it? Could someone who has never seen a painting, sculpture, or sketch still produce meaningful visual art? In a world saturated with cultural influences and artistic traditions, this question becomes challenging to answer. Movements like Outsider Art have already begun to explore the notion that artistic expression can emerge independently of formal training or exposure to traditional art forms. Outsider Art showcases the work of self-taught individuals who, largely disconnected from the art world, create without the influence of established conventions. A more specific subset, Art Brut, focuses on the raw, unfiltered creativity of those entirely outside the established art scene—psychiatric patients, hermits, and spiritualists—people whose art emerges purely from internal drives, uninformed by external artistic influences.

Inspired by these movements, we simulate an "artificial artist" with minimal exposure to art. In this synthetic experiment, we wanted to train a text-to-image model primarily on natural images, with no exposure to visual art. Then adapt the model using a few examples from a specific artistic style to study how well the adapted model can mimic and generalize that style across different contexts.

Powerful text-to-image generators have already proved their ability to produce art, some even winning prestigious competitions (Kuta, 2022). However, their ability is typically attributed to extensive training on large datasets rich with visual art. These models are often so familiar with specific artists' styles that they can replicate them simply by including the artist's name in a text prompt (Heikkilä, 2022). This ease of replication has raised ethical concerns, sparking lawsuits from artists who argue that generative models are imitating their work without permission (Dickstein & Delman, 2023).

In this work, we challenge this paradigm by asking: Can a model with minimal prior exposure to art, but trained on a selected style, compare to these powerful models? Can artistic ability be achieved with just a handful of images, in a controlled manner? To explore this, we develop an art-agnostic model (**Art-Free Diffusion**) that deliberately excludes prior knowledge of visual art. We create an "art-free" dataset (**Art-Free SAM**) using a rigorous filtering method based on both captions and image content to ensure that no artistic elements are included [1]. Then, using the LoRA technique, we introduce the **Art Adapter**, a controlled method of injecting approved artistic knowledge, enabling the Art-Free model to generate art after this carefully authorized input.

Figure 1 shows an example of a generated artwork in the style of an artist Alan Kenny (displayed with his permission). The generated image is conditioned on a text prompt "Guitarist adjusting strings on stage before a performance". This image shows characteristics of Alan's unique style like

---

[1]Our manual inspection shows that the Art-Free dataset may contain 0.05% of paintings and 0.2% of graphic art.

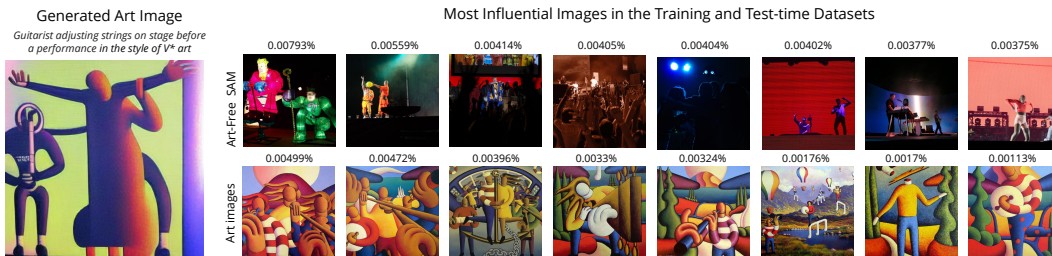

Figure 1: Example of a generated image by a model trained only with real-images (which excludes paintings, drawings, and other forms of graphic art) and adapted using only 11 user provided art images of a target style.

the use of color, smooth boundaries, or geometric shapes, and certainly does not look like a natural image. How can an Art-Free model achieve this after being exposed to just 11 examples of Kenny's work? To answer this question, we apply data attribution method Wang et al. (2023) to analyze which training examples most influenced the generation of the new artistic samples. The top row in the Fig 1 (Right) shows the top influential images within the Art-Free dataset and the bottom row within the images in the Art Adapter training examples. Intriguingly, our analysis reveals that the natural images used in training significantly contribute to the artistic generation process—mirroring the way the natural world shapes real artistic expression.

We conduct extensive experiments to evaluate our approach of art creation with minimal prior artistic knowledge, using measurements of similarity to real art, crowdsourced evaluations of artistic efficacy, data attribution analysis, and an in-depth interview with an artist examining imitations of his own style. Our experiments show that this approach can successfully mimic artistic styles, achieving results comparable to models trained on vast amounts of data.

## 2 RELATED WORK

Text-to-image models have garnered significant attention and popularity, particularly with the advent of open-sourced diffusion models (Song et al., 2020; Song & Ermon, 2019; Rombach et al., 2022a). These models have dramatically improved the quality and fidelity of generated images to user-defined prompts, revolutionizing the field of generative AI. Notably, generated images have not only gained acclaim by winning art competitions (Kuta, 2022), but they have also sparked controversy, leading to lawsuits from artists against companies releasing these models (Guadamuz, 2023). The concerns largely revolve around the models' ability to replicate specific artists' styles (Casper et al., 2023) and their tendency to memorize training data (Somepalli et al., 2023a;b).

In response to these challenges, the computer vision community has proposed several mitigation techniques. Opt-out strategies, allow the removal of specific concepts from model weights (Gandikota et al., 2023a;b; Kumari et al., 2023a; Hong et al., 2024; Lu et al., 2024; Park et al., 2024; Pham et al., 2024; Lyu et al., 2024; Heng & Soh, 2023; Zhang et al., 2024) though these methods often struggle with scalability when dealing with a large number of concepts. It has been also shown that the erased concepts can be re-introduced to the model (Pham et al., 2023). Industry initiatives are also focusing on allowing individuals to opt-out from the training datasets entirely (Spawning AI Team, 2023). However, the effectiveness and implementation of these strategies vary, and they do not fully address the concerns of overfitting and unauthorized style replication. Another line of work involves watermarking training images (Zhao et al., 2023; Min et al., 2024; Cui et al., 2023) to provide traceability and protect intellectual property. While effective to some extent, these approaches differ from our work, which aims to explore the boundaries of what can be introduced into models post-training rather than focusing on in-training safeguards.

In a related vein, Gokaslan et al. (Gokaslan et al., 2024) trained a text-to-image model exclusively on Creative Commons (CC) images. While this approach aims to address ethical concerns by limiting the training data to more permissive licenses, it does not fully resolve the issues, as CC-licensed images can still include artworks that pose similar ethical challenges. In contrast, our work addresses both the ethical concerns raised by artists and the technical challenge of adapting a model that has

been trained exclusively on natural images to learn and generate artistic styles. This exploration offers a distinct perspective on whether such a model can effectively incorporate and reproduce artistic concepts post-training, providing a new avenue for ethical model development.

The most analogous work to ours comes from the field of natural language processing (NLP), where Min et al. (Min et al., 2023) trained a large language model exclusively on a specific subset of data, later introducing an external database for task-specific applications. This approach aligns with our methodology in the sense that both explore the introduction of new data post-training, though our focus is on visual rather than textual data.

Finally, there are also several initiatives dedicated to examining the sources of training data, emphasizing transparency and ethical considerations in the development of AI models (Longpre et al., 2023). Our research contributes to this ongoing discourse by proposing a novel approach to integrating new artistic concepts into models after they have been trained, thereby offering a potential solution to the ethical concerns raised by artists and other stakeholders.

Transferring visual features from one image to another has long been a central topic in computer vision. Image analogies (Hertzmann et al., 2023), for instance, use a pair of example images to demonstrate a desired transformation, which can then be applied to a new image to achieve similar visual effects. Likewise, image quilting (Efros & Freeman, 2023) transfers textures by stitching together small, local patches from a source image, much like assembling a quilt, to synthesize seamless textures on a new canvas. Deep learning methods like Neural Style Transfer (Gatys, 2015) take this further by using convolutional neural networks to extract and recombine deep feature representations of content and style, allowing an image's content to be re-rendered in the artistic style of a reference image. Our method extends beyond traditional texture transfer and image stylization, as we adapt an image generator to a new domain, enabling both the sampling of entirely new images and the stylization of existing ones.

## 3 ART-FREE TEXT-TO-IMAGE DIFFUSION MODEL

**Art-agnostic dataset.** To train an art-agnostic text-to-image model, we require a large text-image dataset that is "art-free". Most commonly used datasets contain numerous examples of art and paintings as diverse visual content is desirable for image generators. We leverage the SAM-LLava-Captions10M dataset (Chen et al., 2023), which is derived from the SA-1B dataset (Kirillov et al., 2023) primarily intended for object segmentation in natural, open-world images. The text captions for SAM dataset are generated by a Large Vision-Language Model (Llava). Prior work has shown that automatically generated captions can also also be effective for training text-to-image models (Chen et al., 2023).

Although the dataset primarily focuses on natural images, we find that it still included instances of visual art, such as stamps, paintings, and other artistic elements. While the dataset may not have been intentionally curated to include artworks, visual art is often difficult to avoid in real-world images. For example, we find photographs of tapestries and baroque architecture featuring artistic details that are "clearly art". Moreover, artistic expression frequently appears in unexpected places, from sculptural designs to logos and branding on everyday objects. Our goal is to distinguish between visual art and natural imagery, ensuring that everyday scenes and objects were represented while minimizing intentional artistic expression. We illustrate in Fig 2 where we draw the line between an art image and not an art image, specifically, we focus on removing graphic arts, and leave other forms of art such as architecture.

To ensure that our training set is free from incidental visual art, we develop a two-stage filtering method. In the first stage, we implement text-based filtering by searching for specific terms in image captions that indicate the presence of visual art. We exclude images whose captions contain keywords such as painting, art, or drawings. In the second stage, we compute a cosine similarity alignment score between each image and a set of art-related terms using the CLIP score (Radford et al., 2021). By manually sampling and ordering images by score for each term, we identify a threshold beyond which the images no longer contained visual art. We refer the reader to Appendix A for further details of the filtering process and the comprehensive keyword list of the art terms. Our resulting **Art-Free SAM dataset**, constructed from SAM-LLava-Captions10M, retains 9,119,455 images after removing 4.7% through text-based filtering and 16.7% through image-based

filtering. We designate 9,140 images as a validation set, yielding a final training dataset of 9,110,315 image-text pairs.

To validate the generalizability of our filtering method, we conducted qualitative manual reviews on both the COCO-2017 and SA-1B datasets. In an initial random sample of 10,000 images from the original SAM dataset, we identified 315 images containing artworks, primarily sculptures, stamps, logos, and paintings. Post-filtering analysis of another 10,000-image sample revealed only 72 images containing artworks, predominantly sculptures. Similar evaluation on the COCO dataset, using a 5,000-image random sample, demonstrated a reduction in art-containing images from 1.06% to 0.12%. Table 2 presents the statistics of samples from both datasets before and after filtering. We will release the filtered Art-Free SAM dataset upon publication.

| | SA-1B$_{ori}$ | SA-1B$_{filtered}$ | COCO$_{ori}$ | COCO$_{filtered}$ |
|---|---|---|---|---|
| #sample | 10,000 | 10,000 | 5,000 | 5,000 |
| paintings | 36 | 5 | 22 | 3 |
| stamp | 71 | 0 | 0 | 0 |
| sculptures | 120 | 52 | 3 | 1 |
| digital art | 14 | 3 | 9 | 1 |
| logo | 36 | 9 | 0 | 0 |
| artwork | 0 | 0 | 10 | 1 |
| sketch | 0 | 0 | 4 | 0 |
| advertisement | 2 | 1 | 5 | 0 |
| drawing | 8 | 0 | 0 | 0 |
| illustration | 4 | 1 | 0 | 0 |
| installation art | 12 | 0 | 0 | 0 |
| mosaic art | 1 | 0 | 0 | 0 |
| tapestry | 3 | 0 | 0 | 0 |
| baroque art | 6 | 0 | 0 | 0 |
| art noveau | 1 | 0 | 0 | 0 |
| pop art | 2 | 0 | 0 | 0 |
| total | 315 ( 3.15%) | 72 ( 0.72%) | 53 ( 1.06%) | 6 ( 0.12%) |

Figure 2: Left: Statistics of artistic images found during manual inspection of the SA-1B and COCO datasets before and after applying the art filter. Right: Example images recognized as art and non-art in our dataset.

**Model architecture.** Our Art-Free Diffusion model is built on a latent diffusion architecture (Rombach et al., 2022b) and has three main modules: the VAE encoder, the UNET, and the Text Encoder. To ensure that no module of our model has been exposed to art, we train both the VAE and UNET from scratch with our Art-Free SAM dataset. The pretrained diffusion models usually use CLIP as the text encoder (Radford et al., 2021; Patashnik et al., 2021), which is trained contrastively to learn associations between images and text. Previous works (Kim et al., 2022) show that a CLIP embedding can manipulate images even in unseen domains. To prevent any art-related knowledge from leaking through the text embeddings, we instead use a language-only Text Encoder based on BERT (Devlin et al., 2019). While the BERT may contain some conceptual knowledge of art, its training process has no access to any visual representations or pixel data containing art, ensuring that the model remains art-free.

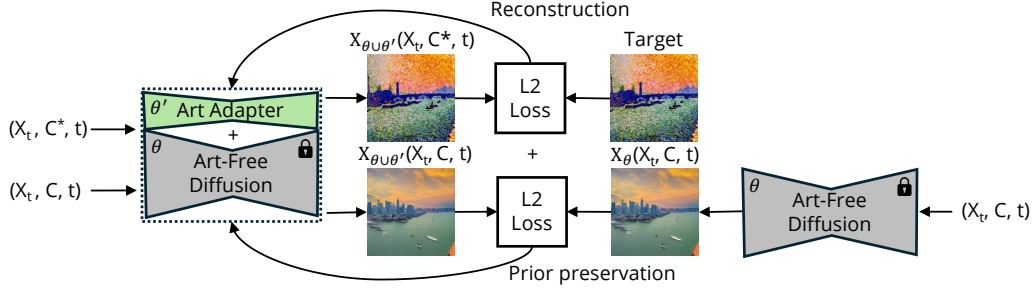

Figure 3: Overview of Art Adapter method. We use pairs of art examples with a caption $C$ describing a content of the image, and a prefix "in the style of V* art" to train the low-rank adapter. Additionally, a prior preservation loss conditioned only on the content caption $C$ is used to preserve the structure and the content information and make a distinction between art and natural images.

## 4 ARTISTIC STYLE ADAPTER

Diffusion models (Ho et al., 2020) represent a class of generative models capable of producing high-quality images by modeling data distributions through successive denoising steps. Intuitively, the forward process incrementally introduces noise to the data, transforming it into Gaussian noise over time. At any given time step, the relationship between the image and the noise can be expressed as:

$$X_t = \sqrt{1 - \beta_t} \cdot X_0 + \beta_t \cdot \epsilon \tag{1}$$

where $X_t$ represents the image at time step $t$, $X_0$ is the original image, $\epsilon$ denotes Gaussian noise with zero mean and unit variance, and $\beta_t$ is an increasing sequence of noise levels. During the reverse process, the model is trained to predict and eliminate the noise $\epsilon$ at each time step to reconstruct the original image. The learning objective can be formulated as:

$$\min_\theta \mathbb{E}\left[\left\|\epsilon_\theta(X_t, C, t) - \epsilon\right\|^2\right] \tag{2}$$

where $\epsilon_\theta$ is the model, $C$ is the condition, which, in our case, is the text prompt. Our model adopts the architecture of the latent diffusion model (Rombach et al., 2022b).

To train an Art-Style Adapter we collect a few examples of artworks in a specific style $X_0 \in \mathcal{A}$ and caption the content of the artwork. This can be done automatically or manually. To connect the newly learned style information with specific tokens in the prompt, we append a text "in the style of V* art" to the content prompt, denoted as $C^*$.

To enable the model to learn this new artistic style, we fine-tune the U-Net module using LoRA (Hu et al., 2021). For a given target artistic image, we define the following reconstruction loss:

$$\mathcal{L}_{\text{recon}} = \left\|\epsilon_{\theta \cup \theta'}(X_t, C^*, t) - \epsilon\right\|^2 \tag{3}$$

Where $\epsilon_{\theta \cup \theta'}$ is the U-Net module with the LoRA updating weights, $t$ is the denoising time step, $X_t$ is the input image at time $t$, and $\epsilon$ is target noise. This loss function helps the model implicitly learn the artistic style and link it to the style modification in the prompt.

$$\mathcal{L}_{\text{preservation}} = \left\|\epsilon_{\theta \cup \theta'}(X_t, C, t) - \epsilon_\theta(X_t, C, t)\right\|^2 \tag{4}$$

Our final loss is $\mathcal{L} = \mathcal{L}_{\text{recon}} + w \cdot \mathcal{L}_{\text{preservation}}$, where $w$ is the hyper-parameter for preservation loss. By combining the reconstruction loss and the preservation loss, our goal is to enhance the model's ability to learn and apply artistic styles while maintaining its ability to generate natural images when no artistic style is specified. This approach enables the model to differentiate between art images and natural images within the same content context, thus enhancing the art style learning. At inference time, we have the ability to fine-tune the balance between style and content in the generated images by carefully selecting the denoising time step at which the art knowledge is injected. This technique allows us to control how much of the artistic style influences the final output. By injecting the style information earlier in the denoising process, the generated image tends to be more stylized, as the model has more time to integrate the artistic features throughout the denoising trajectory. Conversely, injecting the style information at a later stage in the denoising process allows the natural image features to dominate, producing images that retain more natural images with subtle stylistic influences.

## 5 EXPERIMENTS

### 5.1 ART-FREE DIFFUSION

**Model Architecture and Training.** The architecture of our Art-Free Diffusion is based on Stable Diffusion v1.4 (Rombach et al., 2022b). We train the VAE autoencoder from scratch, using a filtered version of the COCO-2017 dataset and a subset of the Art-Free SAM, consisting of 219,439 images. The training is conducted with a batch size of 24, gradient accumulation of 2, and a learning rate of 2e-4, over 15 epochs, which took approximately 16 hours.

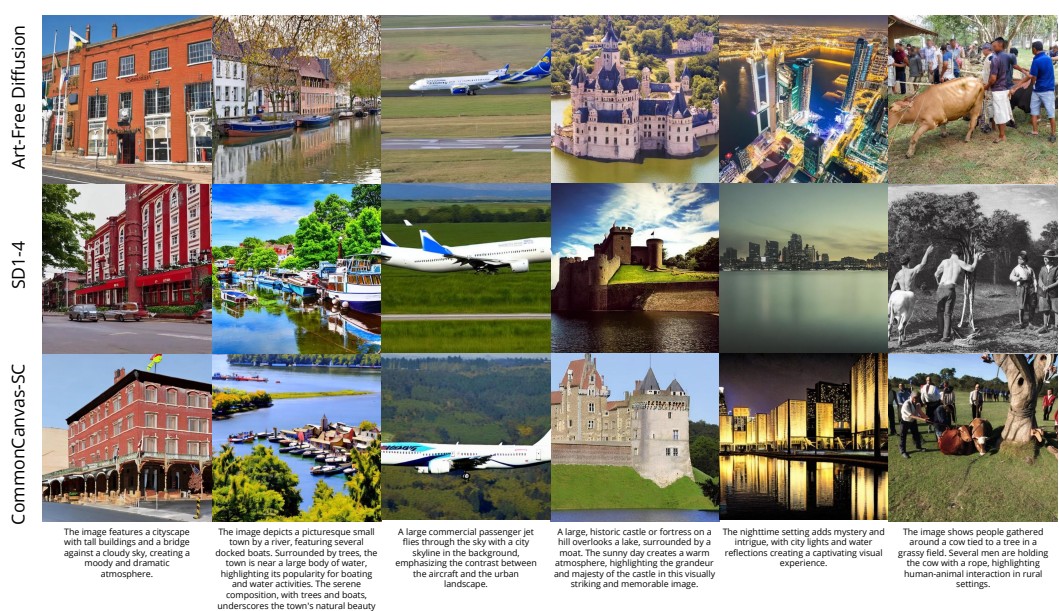

Figure 4: Qualitative comparison across different text-to-image models.

We train the U-Net model on the Art-Free SAM, while keeping the VAE frozen, utilizing a pre-trained BERT base model (uncased) (Devlin et al., 2019) as the Text Encoder. We first train the U-Net under 256 resolution on 7 H100 GPUs, with each GPU using a batch size of 300 and mixed precision of FP16. We apply gradient accumulation of 8 and use a learning rate of 1e-4 with the AdamW optimizer on a 7 H100 GPUs. Then we fine-tune the model under 512 resolution by additional 62,300 steps, using 4 H100 GPUs, learning rate of 5e-5 and batch size of 90, and apply 10% dropping rate with classifier-free guidance sampling (Ho & Salimans, 2022).

**Model Performance Analysis.** We show qualitative comparisons of different models in Fig 4. In Table 1, we compare the performance of three models: CommonCanvas-SC (Gokaslan et al., 2024), Stable Diffusion v1-4, and our Art-Free Diffusion. CommonCanvas-SC employs the same architecture as Stable Diffusion v2 and is trained on 30M commercially sourced samples from the Creative-Commons-licensed (CC) dataset, taken about 73,800 A100 hours. Stable Diffusion v1-4, in its final training stage, utilizes 600M image-text pairs from the LAION-Aesthetics v2 5+ dataset, reported to be trained approximately 200,000 A100 hours (CompVis, 2022). Our Art-Free Diffusion model is trained on approximately 9M images from the Art-Free SAM. We conduct experiments on the test set of the Art-Free SAM (9,140 samples) and 30k samples from COCO-2017.

The evaluation results are presented in Table 1. We observe that all models perform similarly on the Art-Free dataset. However, there is a performance gap when evaluated on the COCO dataset, which can be attributed to several factors. First, the SAM dataset includes blurred faces and license plates to protect the identities of individuals, which may affect performance. Second, the automatically generated captions in the SAM dataset are significantly longer than those in the COCO dataset, introducing a bias toward longer captions. Lastly, our limited resources prevented us from conducting larger-scale training, which impacts our model's competitiveness compared to the other two models. We believe that increasing both the number of images and the training duration would significantly enhance the model's performance.

**Artistic Knowledge Check** In Figure 5, we conduct experiments using prompts that reference famous artworks. The results clearly demonstrate a significant difference between Stable Diffusion v1.4 (SD1.4) and our Art-Free Diffusion. While SD1.4 successfully generates images that closely match the queried artworks, our model produces random images with no recognizable artistic style or elements. This stark contrast highlights the effectiveness of our approach in ensuring that our model possesses no prior knowledge of artworks. Unlike traditional models, which inherently replicate existing artistic styles, our model fundamentally lacks any embedded artistic information.

| Model Name | # Images | Train time (A100 Hours) | Art-Free SAM | | COCO30K | |
|---|---|---|---|---|---|---|
| | | | CLIP ↑ | FID ↓ | CLIP ↑ | FID ↓ |
| **CommonCanvas-SC** | 30M | 73,800 | 0.27 | 13.66 | 0.27 | 8.23 |
| **SD1-4** | 600M | 150,000 | 0.28 | 17.74 | 0.27 | 12.54 |
| **Art-Free Diffusion** | 9M | 5,666 | 0.26 | 12.66 | 0.22 | 24.02 |

Table 1: Model performance comparison between Stable Diffusion v1-4, CommonCanvas-SC, and Our Art-Free Diffusion. Experiments are conducted on the test sets of Art-Free SAM and 30k samples from COCO-2017.

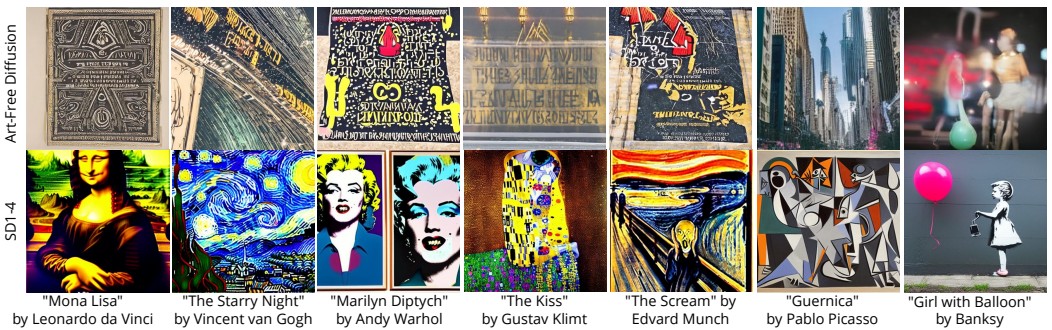

Figure 5: Our model has no prior knowledge of art. It not only fails to generate the artwork indicated by the prompts, but its outputs also lack any apparent stylistic elements.

## 5.2 ART STYLE ADAPTATION

**Implementation Details** For the Art Style Adaptation, we use our Art Adapter with prior preservation weight $w = 50$, and the LoRA rank of 1, incorporating the low-rank Adapters into all attention, linear, and convolution layers (for more details see Appendix B,Appendix C). The learning rate was set to 2e-4 using the AdamW optimizer, and we trained for 1,000 steps with a batch size of 5 and the DDIM noise scheduler. In the experiments, we use 'sks' as the V* token, which serves as a random new token for learning a new art style concept. We select 10 artists and their works from WikiArt to create 10 style sets, each with a distinct style. We manually choose 10 to 40 paintings from each artist with similar color composition, brushstroke techniques, and artistic content to ensure the artistic knowledge dataset has a consistent and coherent style.

To evaluate the art style similarity we use the CSD score (Somepalli et al., 2024), for each sample we compute the mean score between a given generated image and the Art-Adaptation training images. To measure the content fidelity we use the cosine similarity between content features in the generated and original images ($\text{ViT}_c$), further evaluating content consistency or the CLIP score to measure the text and image alignment.

For evaluation, we sample 500 images from the Landscapes HQ(LHQ) dataset (Skorokhodov et al., 2021), with automatically generated text caption. Our experimental setup spans across the 10 style sets. We report the average score over the artists. To validate our results further we conducted a user study on the Amazon Mechanical Turk. We collect pairs of examples showing our Art-Free Diffusion with the Art Adapter for 10 different artists across Image Stylization and Art Generation tasks. Additionally, we test how people perceive real art examples from the same artist. The task displays three reference images demonstrating the style of an artist and a pair of examples. The user's task is to choose which of the two images is more similar in style to the reference images. We include a test in the user study, that we then use to remove unreliable participants. In total we collect 1700 answers across 16 different users. We discuss the results below.

**Image stylization** We evaluate our method on an image stylization task, transforming image styles while preserving content, using the LHQ dataset. Comparisons are made against SD1.4 baselines: SD1.4 (Adapter), which uses the learned Art-Style Adapter with a new text token; SD1.4 (Text), which queries the model using the artist's name; and SD1.4 (Adapter + Text), combining both. Since our model lacks direct knowledge of artist names, text guidance has limited impact, as shown in our quantitative results. For these methods, we apply DDIM inversion to noise a real image to

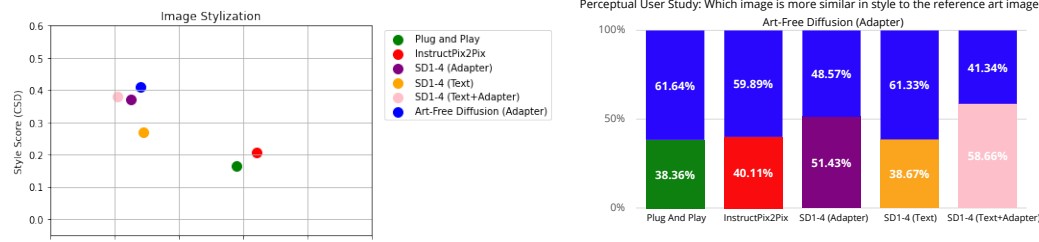

Figure 6: Art-Free Diffusion (Adapter) changes image style more than editing baselines, yielding higher style scores, lower content scores, and overall performance similar to SD1.4 (Adapter).

Figure 7: Results of the Perceptual User Study; Art-Free Diffusion (Adapter) method is preferred over image editing baselines and favored less with SD1.4 (Adapter), however the margin of preference is narrow between the baselines.

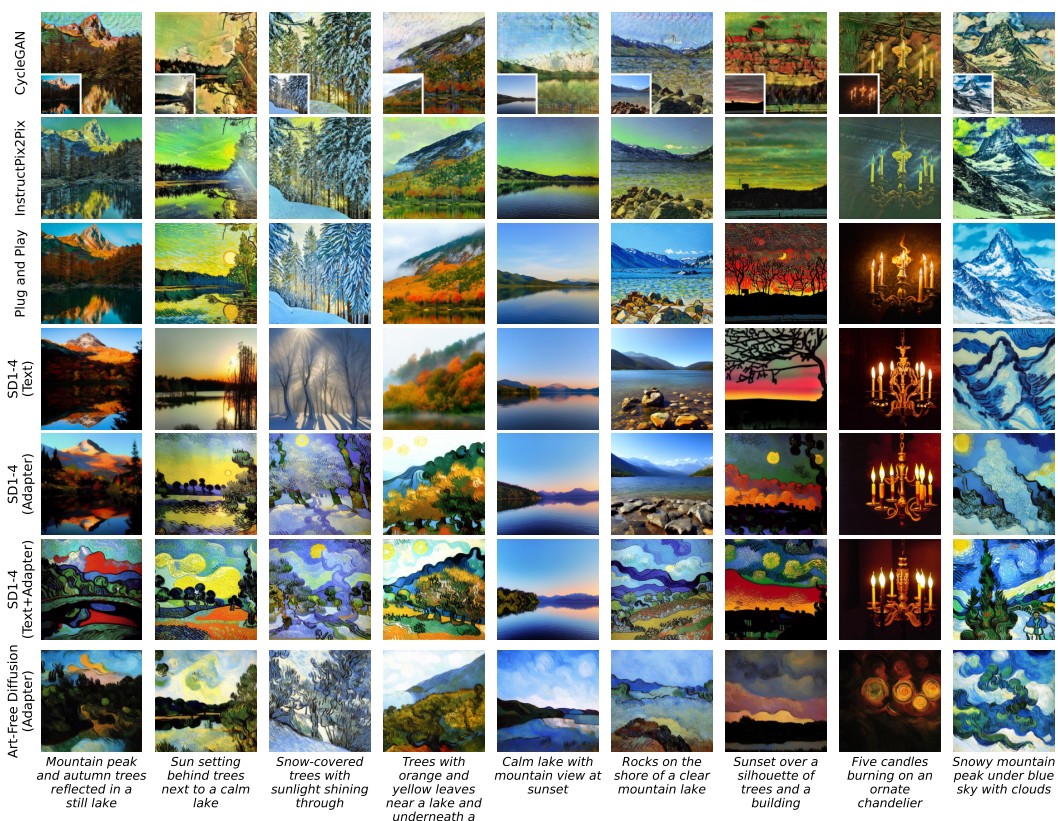

Figure 8: Comparion of our method and other image stylization baselines for the artist Van Gogh. All captions contain a suffix "in the style of Vincent van Gogh", Our model and SD1.4 + Art Adaptor are prompted with suffix "in the style of V* art".

step 800, then denoise while changing the text prompt and applying the adapter where needed. We also compare against Plug and Play (Tumanyan et al., 2023), which edits internal model features by appending "a painting by [artist]" to the caption, and InstructPix2Pix (Brooks et al., 2023) using the prompt "turn into a [artist] painting." We also include qualitative comparison with CycleGAN (Zhu et al., 2017) for Monet and Van Gogh.

Qualitative results for Van Gogh are shown in Fig.8, with quantitative comparisons in Fig 6. Plug and Play and InstructPix2Pix yield slight changes to the original image as reflected in higher content scores and lower style scores. Stable Diffusion (Text) achieves a high style score, indicating strong model knowledge, while the Art Adapter improves results in all cases. The perceptual user study on style is comparable with our automatic evaluation (Fig 7), participants generally preferred our method over all baselines except Stable Diffusion with an Art Adapter. When the Art Adapter has no extra style information beyond the Art Examples, users favored our method 48.57% of the time. Remarkably, when participants were shown real art examples, they found our images more similar to the artistic style 47.6% of the time, compared to 31.8% for Stable Diffusion with Adapter and 23% for Plug and Play-edited images.

**Art Generation** We address the task of Art Generation, focusing on creating images in a specific artistic style. Stable Diffusion, known for its ability to replicate styles by simply prompting with artist names, serves as a baseline due to its extensive training on artworks. We compare this to our model in 5.2 (Right). For this evaluation we use a set of 500 captions from the LHQ dataset for all the artists and a set of 20 automatically generated captions that relate more to the concepts from the Art Examples (Custom Captions), we generate images with both Art-Free Diffusion (Adapters) across 10 artists. Art-Free Diffusion (Adapter) outperforms SD1-4 (Text) in style, with CSD scores of 0.55 and 0.47 on Custom Captions and LHQ, respectively, compared to 0.45 and 0.34 for SD1-4 (Text). However, it shows slightly lower content preservation, scoring 0.20 and 0.19 versus 0.26 and 0.22. The balance between style and content can be fine-tuned by applying the Art Adapter at later diffusion stages (see Appendix E), though in this experiment, it is applied throughout. Notably, Custom Captions yield higher style scores than LHQ for both models, with qualitative examples presented in Fig. 9.

Our perceptual user study supports the style score findings, with participants preferring our method over SD1.4 (Text) 63% of the time. Although users were tasked with selecting the image most similar in style to real artworks, there was notable confusion. Participants chose images generated by our method 38.8% of the time and those from SD1.4 (Text) 32% of the time, indicating that generated images were often mistaken for real art in terms of style.

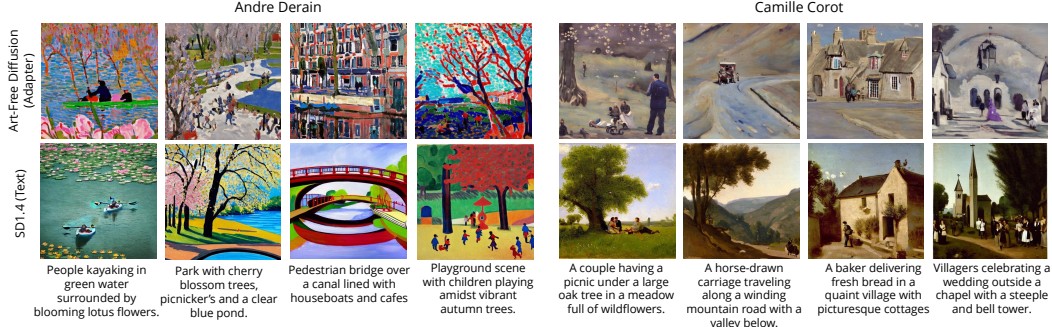

Figure 9: Comparison of Art-Free Diffusion art generation with directly generating art images with Stable Diffusion 1.4.

**Data Attribution** We find that our Art Adapter can generalize from a small Art-Style training set and generate seemingly novel images that are coherent with the given artistic style. To better understand which training images contributed to the synthesized image, and to check whether the art filtering may have overlooked some art content that influenced the result, we applied the data attribution technique proposed by (Wang et al., 2023). The results of this experiment are shown in Fig. 10. For each generated image, we retrieved the top eight relevant images from both the Art-Free SAM and Art-Style examples, separately. Since we explicitly expect an image in a certain style, it is natural that the art images are ranked highly, as seen in Fig. 10 a),d). However, this method also enables us to trace the influence of art-free training examples. For instance, in Fig. 10 b) and c), despite the very abstract and distinctive style, we can observe which photographs of the natural world contributed most to the generated image. Similarly, in Fig. 10 e), the generated image shows a forest with a distinctive color palette and brushstrokes characteristic of Hokusai's style, yet much of the content is strongly attributed to the art-free images.

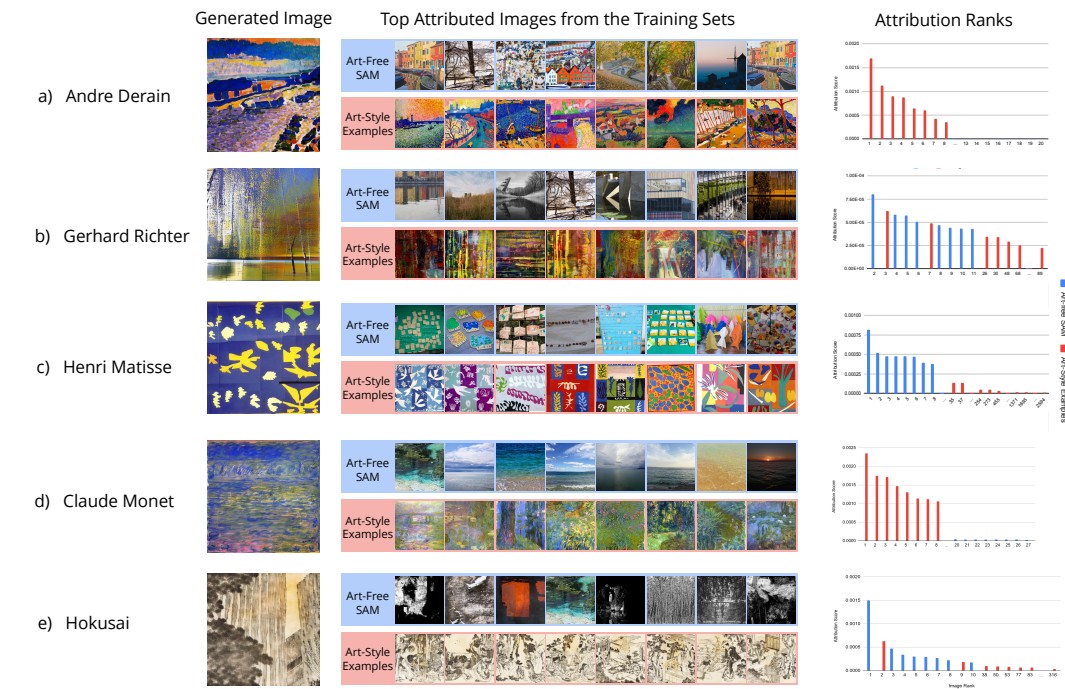

Figure 10: Results from our data attribution experiments on synthesized art images. While the generated images reflect the distinct artistic styles of each artist, the training images that contributed the most came from both the Art-Free dataset and the Art-Style examples.

**Introducing the Art Adapter to the Artist** To explore the artistic community's reaction to AI-generated art, we conduct an interview with the renowned artist Alan Kenny. Upon obtaining Alan's permission, we train an Art Adapter on 11 artworks showing his distinctive style. We describe our work and present Kenny with the generated images imitating his style.

In the interview, the artist expresses a blend of astonishment and familiarity when observing the AI-generated art, remarking, "I didn't expect [this quality] if you were using a base model of blank canvas... you probably achieved more than I would have expected for a base model with no information." He acknowledges that the AI has captured aspects of his distinct style to the extent that, "if you were to post some of these images online, I would get people texting me, 'I see your images.' They would spot it, and I spot it." Despite noting that "compositionally, it is weak" and contrasting this with his own "well thought and meticulous" compositions, he recognizes that "there are some very positive things" in the AI's work.

The artist describes the experience as "terrifying and a bit exciting at the same time," specifically pointing out how the AI imitates his signature "gradation of the landscape" and "gradation of the shapes." Though he felt his style is largely captured, he admits, "there is kind of originality to them... I see me in them, yes, very strongly... but there is an originality to some of the images."

## 6 DISCUSSION

In this paper, we introduce the Art-Free Diffusion model, which explores the ability to mimic an artistic style with minimal exposure to art. We propose a simple method for training an Art Adaptor to achieve this goal and evaluate its performance in image stylization and art generation tasks using both automatic metrics and a perceptual user study. Our experiments demonstrate that the model can successfully imitate artistic styles. Additionally, we consulted a professional artist to gather expert feedback on how well the artificial model replicates his artistic style, further validating our findings. To support our thesis, we applied a data attribution method to understand how a model with limited knowledge of artistic styles can still produce artistic images. The results provide intuitive insights into how the natural world can influence and inspire art.

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

SUPPLEMENTARY MATERIAL

# A    ARTWORK FILTERING METHODOLOGY

Our artwork filtering process operates on both image and caption levels to ensure comprehensive coverage. For image-level filtering, we define a set of concepts to be excluded:

> painting, art, artwork, drawing, sketch, illustration, sculpture, stamp, advertisement, logo, installation art, printmaking art, digital art, conceptual art, mosaic art, tapestry, abstract art, realism art, surrealism art, impressionism art, expressionism art, cubism art, minimalism art, baroque art, rococo art, pop art, art nouveau, art deco, futurism art, dadaism art

Figure 11 presents a histogram of CLIP scores for images associated with the word "painting" in their captions. This distribution is derived from a subset of the SA-1B dataset, comprising 11,186 images (1/1000th of the complete SA-1B dataset).

For caption-level filtering, we exclude the following terms (case-insensitive):

> painting, paintings, art, artwork, drawings, sketch, sketches, illustration, illustrations, sculpture, sculptures, stamp, stamps, advertisement, advertisements, logo, logos, installation, printmaking, digital art, conceptual art, mosaic, tapestry, abstract, realism, surrealism, impressionism, expressionism, cubism, minimalism, baroque, rococo, pop art, art nouveau, art deco, futurism, dadaism

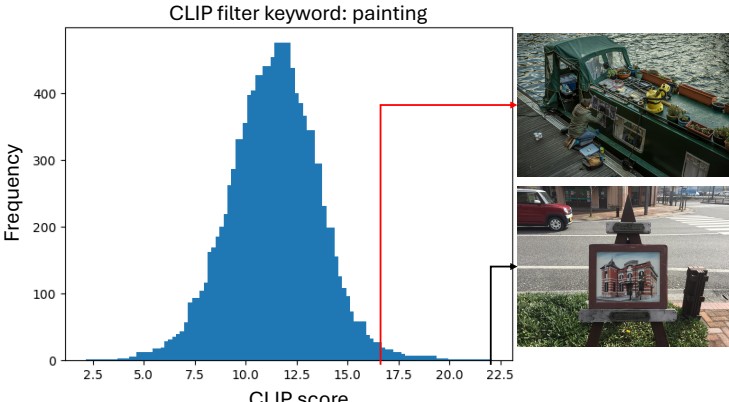

Figure 11: Histogram of the CLIP score of images with the word "painting" in the caption. The distribution shown is from a subset of the SA-1B dataset. The red line represents the filtering threshold (17) we selected. Our strict threshold aims filters out all the art, even incidental art like a picture of a man painting.

# B    LoRA RANK ANALYSIS

We conducted an analysis to determine the effect of LoRA rank on the art adapter's performance. Table 2 presents the results of our model with LoRA ranks ranging from 1 to 64. Our findings indicate that LoRA rank does not significantly impact model performance. Consequently, we conclude that setting the LoRA rank to 1 is sufficient for our experiments, which helps to optimize resource utilization.

| Model | Text Guid. | Art Adaptor | LoRA Rank | CSD↑ | LPIPS↓ | ViTc↑ | CLIPc↑ |
|-------|-----------|-------------|-----------|------|--------|-------|--------|
| OURS | × | ✓ | 1 | 0.41 | 0.63 | 0.28 | 0.19 |
| OURS | × | ✓ | 64 | 0.36 | 0.61 | 0.29 | 0.20 |

Table 2: Rank analysis of LoRA on style transfer ability. We find that the rank of LoRA does not improve the model learning performance, even when changed from 1 to 64.

## C  PRIOR PRESERVATION ANALYSIS

We investigated the influence of the prior preservation loss weight ($w$) in the art adapter across different models. Table 3 presents the results for Stable Diffusion and our model with varying $w$ values.

For our model, the prior preservation loss substantially enhances learning performance, with CSD increasing from 0.28 to 0.41 when $w$ is set to 50. This demonstrates that the preservation loss effectively aids the model in distinguishing between art images and natural images. The effect remains robust across different weight values, with performance remaining nearly constant when $w$ is set to 20 or 100 (up to 0.03 difference in CSD).

| Model | Text Guid. | Art Adaptor Rec. | Prior$_w$ | CSD↑ | LPIPS↓ | ViTc↑ | CLIPc↑ |
|-------|-----------|------------------|-----------|------|--------|-------|--------|
| OURS | × | ✓ | 0 | 0.28 | 0.59 | 0.32 | 0.19 |
| OURS | × | ✓ | 20 | 0.39 | 0.63 | 0.27 | 0.18 |
| OURS | × | ✓ | 50 | 0.41 | 0.63 | 0.28 | 0.19 |
| OURS | × | ✓ | 100 | 0.38 | 0.61 | 0.29 | 0.20 |

Table 3: Analysis of prior preservation loss weight ($w$) on our model. Experiments are conducted on style transfer, with noise added at the 800th time step.

## D  ART-AGNOSTIC MODEL VERIFICATION

To verify the art-agnostic nature of our model, we conducted a textual inversion experiment as suggested by Pham et al. (2023). Figure 12 illustrates that our model fails to produce the target style using textual inversion, further confirming its lack of prior artistic knowledge.

Textual Inversion - van Gogh

Our model  SD1-4

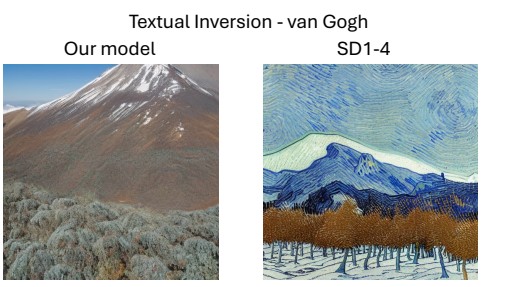

Snow-covered mountain peak behind a field of leafless brown bushes in the style of V* art

Figure 12: Through textual inversion using paintings by van Gogh, we found that, unlike SD1-4, our model cannot generate images in the corresponding style. This indicates that our model cannot be hacked to generate artwork through prompt space searching, demonstrating it has no prior knowledge of art.

### D.1  MODEL EDITING AND CONTROLLING ABILITY

Despite being trained on a significantly smaller and less diverse dataset limited to natural images, our art-agnostic model demonstrates comparable editing and control capabilities to competitive models. This is evident in both single-image editing and customization experiments.

In Figure 13, we qualitatively illustrate the single-image editing process using the Plug-and-Play method (Tumanyan et al., 2023) applied to our model. We provide editing examples on both real and generated images, demonstrating the model's ability to replace a pyramid with a large mountain, both with and without the artistic adapter (weight 1.5) of van Gogh.

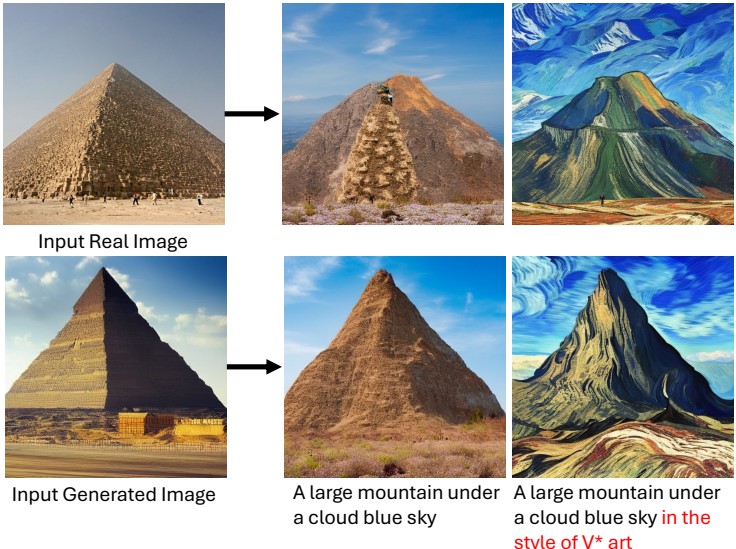

Figure 13: Plug-and-Play editing on our model. We provide both editing on real and generated image examples. We replace a pyramid to a large mountain both without and with the artistic adaptor of van Gogh.

Furthermore, we demonstrate our model's customization abilities using the Dreambooth technique (Ruiz et al., 2023). We learned the concept of a barn using 7 training images from the Custom-Concept101 dataset (Kumari et al., 2023b). The model was trained to generate the barn in various contexts, utilizing 200 prior samples from Stable Diffusion v1-4, with a prior preservation loss of 1.0, a learning rate of 5e-6, and 250 training steps on 2 GPUs.

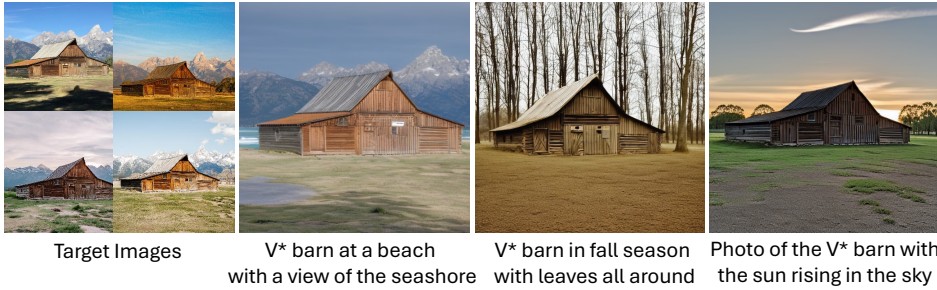

Figure 14: Dreambooth editing on our model. We send 7 barn example images to the model and ask it to generate the barn in various contexts.

# E ADAPTER TIME STEP ANALYSIS

We analyzed the effect of the adapter time step on art generation results. Figure 15 shows the art generation outcomes with different adapter time steps. Intuitively, the model generates more style information when the adapter starts earlier (left) and more content information when the adapter starts later (right).

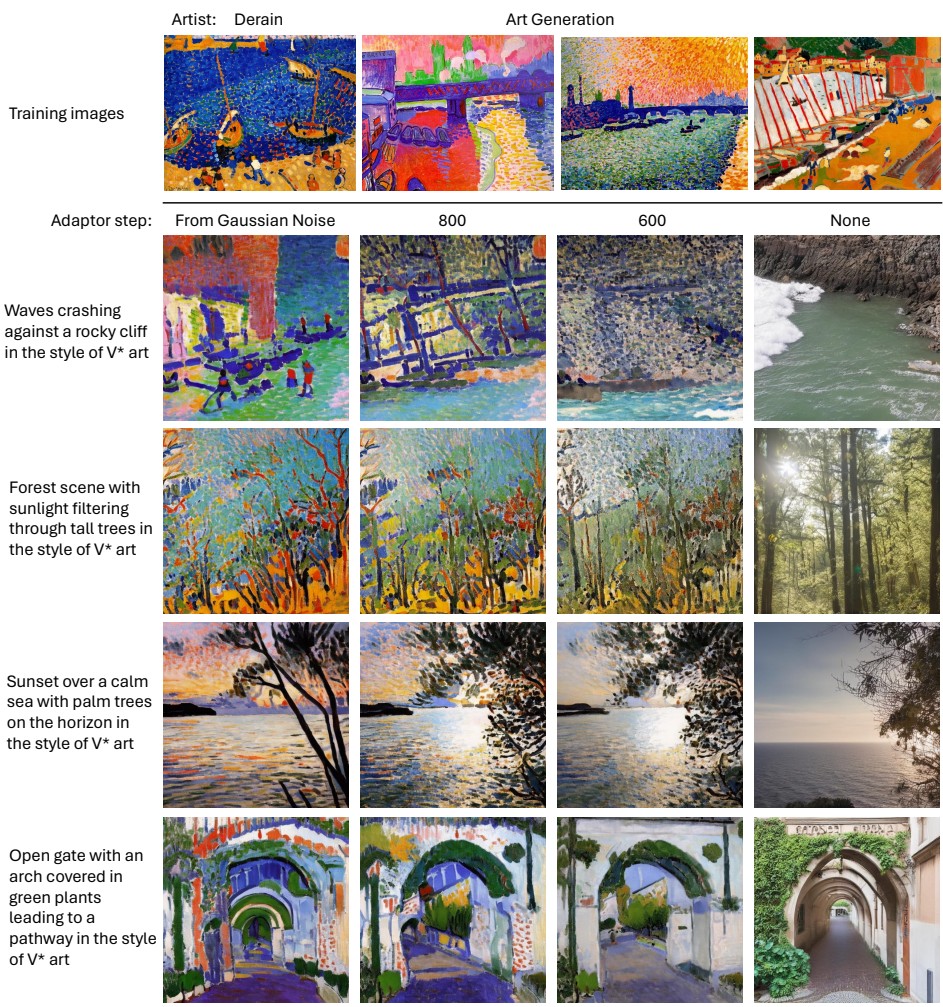

Figure 15: Art generation results with different adaptor time steps. The larger the time step, the earlier the adaptor starts, the more style information is generated. "From Gaussian Noise" means the art adaptor is used in the whole generation process.

# F ADDITIONAL ART GENERATION RESULTS

We present additional art generation results (including both art generation and image stylization) and examples of the training images in Figures 16–25. These results demonstrate our model's capabilities across various artistic styles, including Corot, Derain, Hokusai, Klimt, Matisse, Monet, Picasso, Richter, van Gogh, and Warhol.

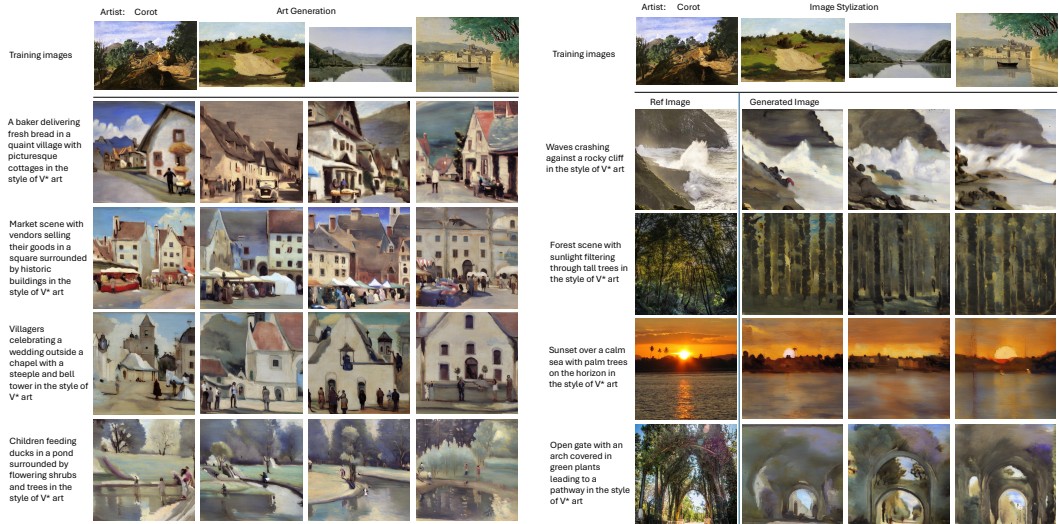

Figure 16: Qualitative results of learning the artistic style of Camille Corot. (Left) Art Generation, (Right) Image Stylization.

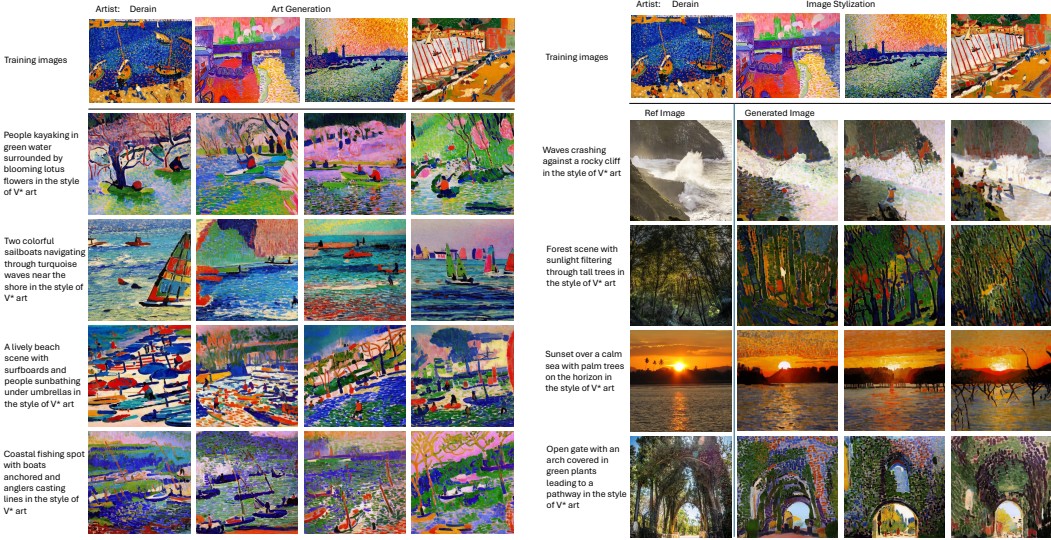

Figure 17: Qualitative results of learning the artistic style of André Derain. (Left) Art Generation, (Right) Image Stylization.

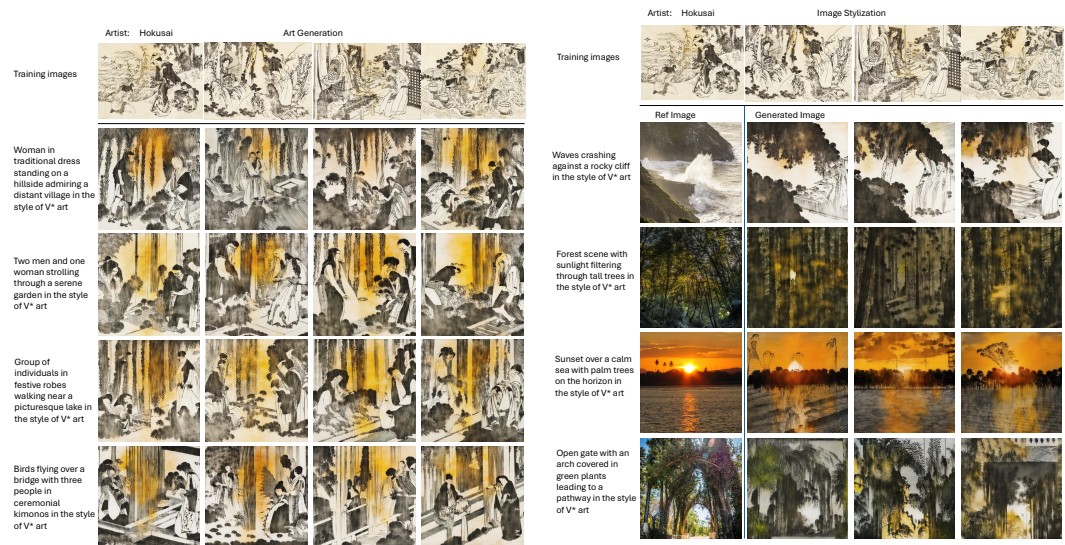

Figure 18: Qualitative results of learning the artistic style of Hokusai. (Left) Art Generation, (Right) Image Stylization.

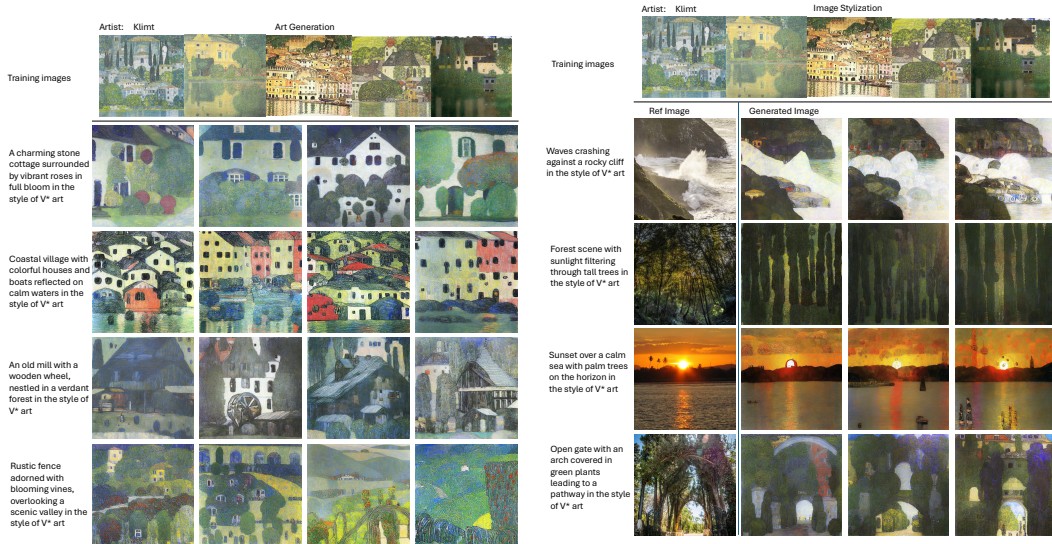

Figure 19: Qualitative results of learning the artistic style of Gustav Klimt. (Left) Art Generation, (Right) Image Stylization.

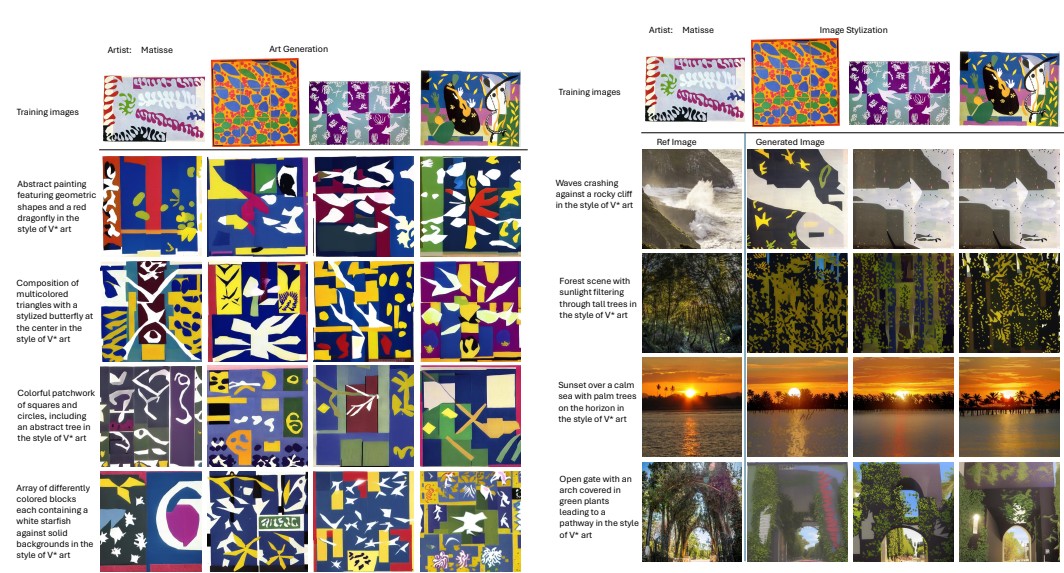

Figure 20: Qualitative results of learning the artistic style of Henri Matisse. (Left) Art Generation, (Right) Image Stylization.

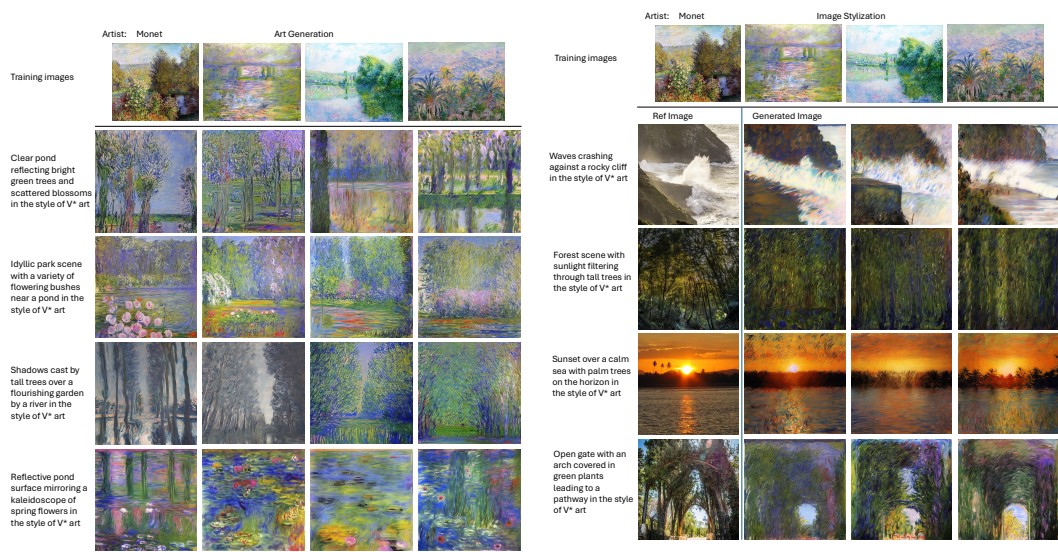

Figure 21: Qualitative results of learning the artistic style of Claude Monet. (Left) Art Generation, (Right) Image Stylization.

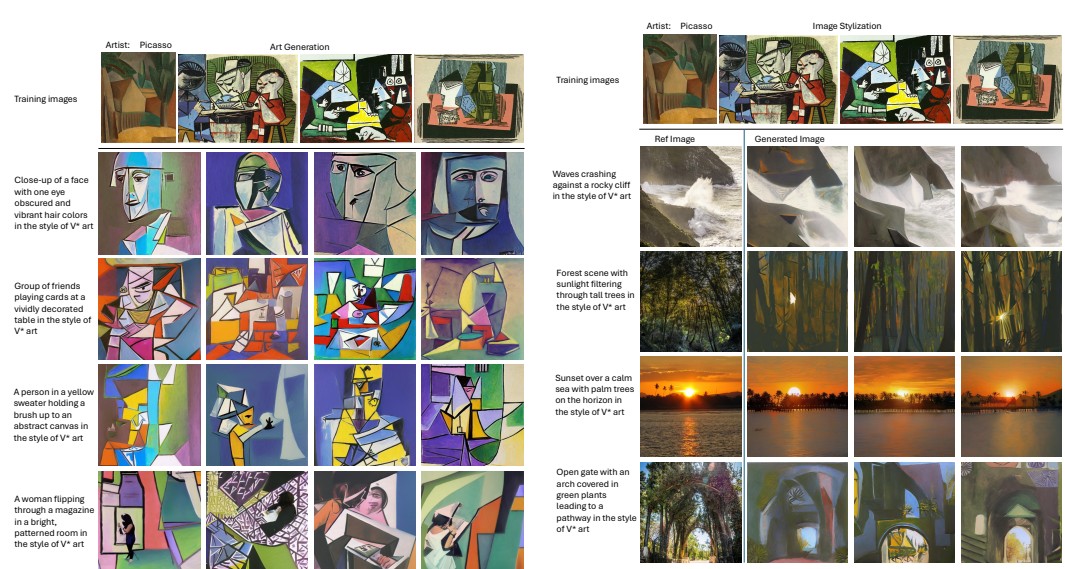

Figure 22: Qualitative results of learning the artistic style of Pablo Picasso. (Left) Art Generation, (Right) Image Stylization.

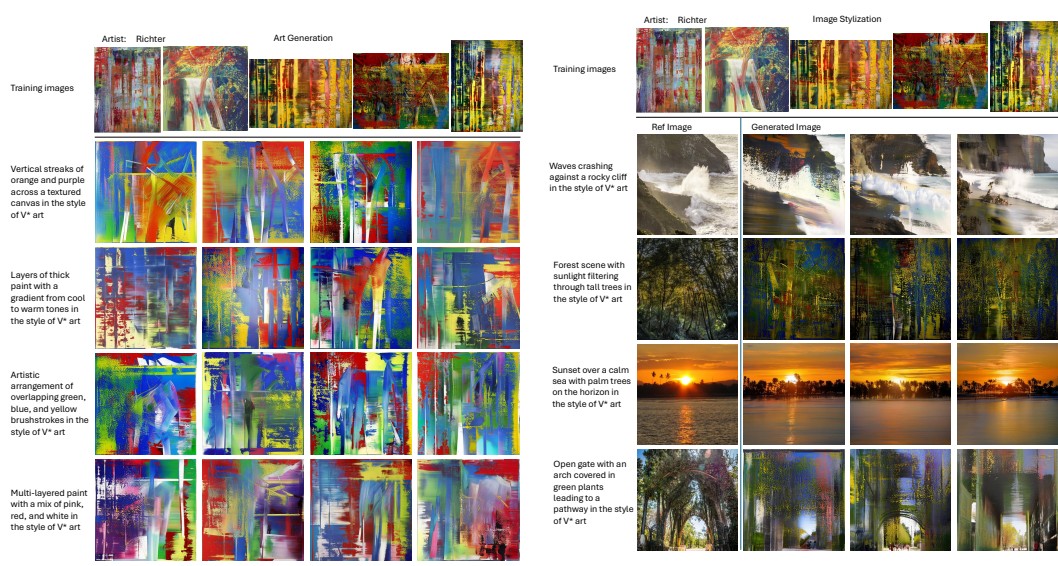

Figure 23: Qualitative results of learning the artistic style of Gerhard Richter. (Left) Art Generation, (Right) Image Stylization.

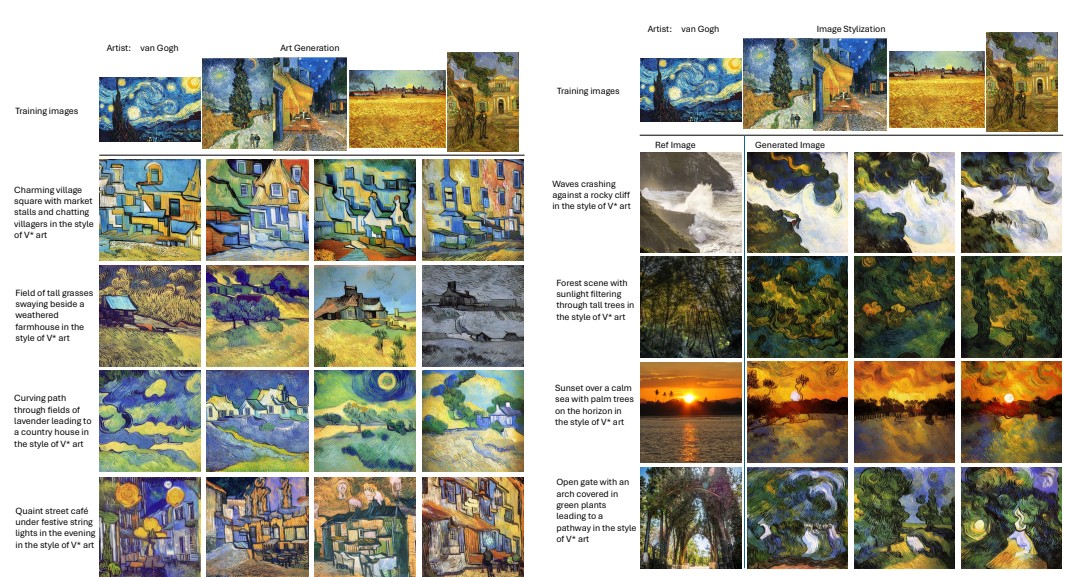

Figure 24: Qualitative results of learning the artistic style of Vincent van Gogh. (Left) Art Generation, (Right) Image Stylization.

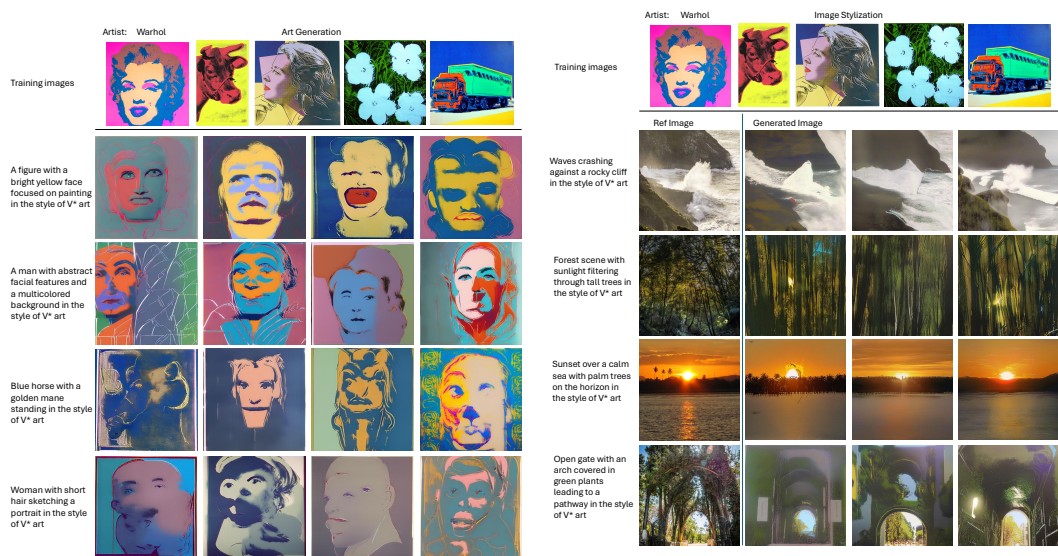

Figure 25: Qualitative results of learning the artistic style of Andy Warhol. (Left) Art Generation, (Right) Image Stylization.

