# OpenReview forum: "Art-Free Generative Models: Exploring Art Creation Without Prior Artistic Knowledge"
_ICLR.cc/2025/Conference — ICLR 2025 Conference Withdrawn Submission_

### Official Review · Reviewer_Lbp3 · 2024-10-24

**Soundness:** 3
**Presentation:** 3
**Contribution:** 2
**Rating:** 5
**Confidence:** 3

**Summary:**

This work explores the question: is exposure to art necessary for creating it? It designs a synthetic experiment to investigate this question. It trains a text2image model on natural images to avoid exposure to visual art, and then adapts the model using a few examples from a specific artistic style to study how well the adapted model can mimic and generalize that style across different contexts.

**Strengths:**

1.	The problem proposed by this work is important. On the one hand, it tries to answer where text2image model performs like human, where her/his art may emerge from internal drives rather than external artistic influences. On the other hand, it attempts to solve the ethical concerns whether the generative models are imitating artists’ work without permission.

2.	The experiment effectively answers the proposed questions. It is simple and provides enough details for reproduction.

**Weaknesses:**

1.	It is expected that there will be in-depth discussion about more types of art instead of only the art from WikiArt. For example, digital arts (refer to [1]) like icons, flat designs, or cartoons are also widely perceived and of great value in modern life. This kind of art is different to the art from WikiArt in the sense that it is more abstract, e.g., using several circles and arcs to represent a happy face. Can the text2image model trained on natural images also be adapted to generate other types of art?

2.	The study for other style transfer methods is missed (see [1],[2] and [3]). Can these methods also adapt the text2image model trained on natural images to perform art generation? Is there any performance difference between existing methods and the proposed style adaptor? If there is no difference, why should we propose a new method? If there is difference, what is it and what makes the proposed method different?

[1] StyleDrop: Text-to-Image Generation in Any Style

[2] InstantStyle: Free Lunch towards Style-Preserving in Text-to-Image Generation

[3] DEADiff: An Efficient Stylization Diffusion Model with Disentangled Representations

**Questions:**

In line 377, why is DDIM inversion used for the proposed method since it is already trained with the awareness of “in the style of V* art”? Is it necessary?

---

### Official Review · Reviewer_Su3h · 2024-10-29

**Soundness:** 3
**Presentation:** 3
**Contribution:** 2
**Rating:** 5
**Confidence:** 4

**Summary:**

In this paper,the authors explore an interesting question,for a text-to-image model that has not seen any artwork, i.e., trained only on natural images, how much art-related knowledge is needed for the model to generate images in an artistic style.To dive into this question,the authors trained an art-free diffusion model on an art-free dataset, and finetune the model with a few images using a lora,experiments show that this approach can successfully mimic artisticstyles, achieving results comparable to models trained on vast amounts of data.

**Strengths:**

1.The article is written in a smooth and highly understandable manner.
2.The article conducted a thorough and in-depth analysis of various datasets，proposes a two-stage dataset construction method, curated an art-free dataset, and the authors promise to make it open source.
3.The article conducted extensive experiments, and the results are convincing.

**Weaknesses:**

1.While the perspective of exploring the problem is quite new, this paper essentially just trains a LoRA model, which is a very common practice in the area of Generative models, so the novelty and technical contribution of this paper is weak.
2.Some comparison experiments in this paper is unfair.The model in this paper is a specialized model fine-tuned with LoRA, while SD is a general model pre-trained on a large number of artistic works. Comparing these two might be a bit unfair.

**Questions:**

1.Generally，this paper's experiments is comprehensive and well-designed, but consider the quality and the bar of ICLR, i want the author
to provide more technical contribution and deeper analysis of the question, like what is the inner connection between natural image and artist work.
2.I wander why the authors choose to use SAM-1B as the base dataset, the picture in this dataset is relatively complex and have many different elements, which may pose challenge to filter the art-style images, and the filter method is simple, so I suspect wheather the filtered dataset is really "art-free".

---

### Official Review · Reviewer_tpn3 · 2024-10-31

**Soundness:** 3
**Presentation:** 3
**Contribution:** 2
**Rating:** 3
**Confidence:** 4

**Summary:**

This paper examines how much prior exposure to art-related data is necessary for generating art through diffusion models. By training a text-to-image model without any (or very small) art content, the study introduces a minimalist approach using a few artistic examples to guide style adaptation.

**Strengths:**

1. **Motivation**: The paper addresses a unique and relevant question regarding the necessity of art-related datasets in AI-generated art. This motivation is timely and important, especially in light of recent copyright concerns in AI-driven art.
2. **Dataset**: The authors provide a well-curated art-free dataset, a valuable resource for researchers in generative art who seek to avoid style replication while maintaining creative quality.
3. **Potential Impact**: Although there are limitations (discussed below), the study offers meaningful insights for the art and multimedia research communities. By showing how non-art datasets can still yield creative outputs, it broadens the conversation on dataset dependencies in creative AI.

**Weaknesses:**

1. **Unclear Impact**: The study’s overall impact and objectives lack clarity, particularly in terms of practical application. If the paper’s main aim is to prevent style replication, it’s uncertain if users would accept potential quality trade-offs in actual art creation workflows. Additional user studies or interviews could help substantiate the motivation. For example, the authors could explore artists’ willingness to adopt the art-free diffusion model after clearly explaining its strengths and weaknesses compared to more conventional models like Midjourney or Stable Diffusion. Understanding the extent to which artists tolerate performance loss would also be valuable—for example, if the model is used early in the creative process to spark inspiration, style fidelity or image quality may be less critical. However, if it’s applied in the later stages of art creation, aesthetic quality may be crucial, which could limit the applicability of the art-free model.
2. **Evaluation**: The evaluation relies heavily on user studies focused on Van Gogh’s style (Figure 9), making it challenging to generalize the results across diverse artistic styles. Broader evidence is needed to support the model’s effectiveness across a range of art genres. To strengthen the results, additional experiments across diverse art genres or styles, such as illustration, comics, and others, would be valuable.
3. **Novelty**: While the dataset curation is commendable, the model and experimental approach are relatively standard, which may feel incremental for an ICLR submission. The way the authors train or fine-tune the art-free diffusion model is very similar to the usual latent diffusion model and DreamBooth+LoRA fine-tuning process. Nevertheless, the study retains relevance for art and multimedia research.
4. **Artist Feedback**: The artist interviews primarily address general attitudes toward AI in the art community rather than the specifics of the art-free diffusion approach. Conducting in-depth interviews comparing traditional and art-free models could provide stronger insights into the study’s implications.

While my review may seem largely critical, I recognize the unique contributions and potential value of this research within its niche.

**Questions:**

Could you clarify the implications of the data attribution experiments?
It seems intuitive that, with an art-free dataset for pre-training and art examples for few-shot tuning, content attributes would align with the art-free dataset, while stylistic elements align with the art examples. Does this experiment reveal a deeper significance?

---

### Official Review · Reviewer_kAPL · 2024-11-01

**Soundness:** 3
**Presentation:** 4
**Contribution:** 3
**Rating:** 5
**Confidence:** 2

**Summary:**

The paper presents an innovative exploration into the question of how much prior art knowledge is necessary to create art. The authors introduce a novel text-to-image generation model, the Art-Free Diffusion model, which is trained without access to art-related content. They propose a straightforward yet effective method to train an art adaptor using only a few examples of selected artistic styles. The experiments conducted demonstrate that the art generated by this method is perceived by users as comparable to that produced by models trained on extensive, art-rich datasets. This work provides valuable insights of the natural world data on art creation.

**Strengths:**

- The paper is very well-written, with clear expression and a logical structure.
- The experiments are thorough, with different comparative experiments designed for various evaluation aspects.
- The approach is quite novel: it explores whether a diffusion model trained without art images in the training set can still generate artistic images.

**Weaknesses:**

- **Writing**:
  - Figure 3 is not referenced.
  - The resolution of Figure 6 is too low.
  - Typo: mixed usage of Art Adapter and Art Adaptor

- **Experiments**:
  - One of the main comparison models, SD1.4, is relatively old. Why not compare with the latest models such as SDXL or SD3 ? The COCO dataset is not an art dataset.
  - **Model Performance Analysis**:
 From the results in Table 1, it appears that the performance of the Art-free diffusion model on this dataset is still not as good as that of other models.
  - **Image Stylization**: The main comparisons are with Plug and Play, InstructPix2Pix, and CycleGAN. However, there is no comparison with more recent works such as StyleID.

- **Analysis**:
  - In Lines 73-74, the authors claim that "our analysis reveals that the natural images used in training significantly contribute to the artistic generation process," but this point is not supported by quantitative data. Additionally, in the qualitative analysis in Figure 10, Rows (a, d) do not support this conclusion.

**Questions:**

- The main objective of this paper is to obtain an art-agnostic diffusion model by removing art data during the training phase, and then to learn specific art styles by fine-tuning a style adapter. I do not quite understand the advantage of this two-stage approach compared to the standard method of training a model with art images. It seems to simply split the original single-stage training into two steps, each using different types of data (non-art and art).
- As mentioned above, it would be beneficial to compare the proposed method with some of the latest approaches in the field.

---

### Note · Authors · 2024-11-15

I have read and agree with the venue's withdrawal policy on behalf of myself and my co-authors.